# Circulating miRNAs Expression in Myalgic Encephalomyelitis/Chronic Fatigue Syndrome

**DOI:** 10.3390/ijms241310582

**Published:** 2023-06-24

**Authors:** Irene Soffritti, Sabine Gravelsina, Maria D’Accolti, Francesca Bini, Eleonora Mazziga, Anda Vilmane, Santa Rasa-Dzelzkaleja, Zaiga Nora-Krukle, Angelika Krumina, Modra Murovska, Elisabetta Caselli

**Affiliations:** 1Department of Chemical, Pharmaceutical and Agricultural Sciences, and LTTA, University of Ferrara, 44121 Ferrara, Italy; irene.soffritti@unife.it (I.S.); maria.daccolti@unife.it (M.D.); francesca.bini@unife.it (F.B.); eleonora.mazziga@unife.it (E.M.); 2Institute of Microbiology and Virology, Rīga Stradiņš University, LV-1067 Riga, Latvia; anda.vilmane@rsu.lv (A.V.); santa.rasa-dzelzkaleja@rsu.lv (S.R.-D.); zaiga.nora@rsu.lv (Z.N.-K.); modra.murovska@rsu.lv (M.M.); 3Faculty of Medicine, Department of Infectology, Rīga Stradiņš University, LV-1006 Riga, Latvia; angelika.krumina@rsu.lv

**Keywords:** myalgic encephalomyelitis, chronic fatigue syndrome, microRNA, HHV-6A, HHV-6B, biomarkers

## Abstract

Myalgic encephalomyelitis/chronic fatigue syndrome (ME/CFS) is a complex multifactorial disease that causes increasing morbidity worldwide, and many individuals with ME/CFS symptoms remain undiagnosed due to the lack of diagnostic biomarkers. Its etiology is still unknown, but increasing evidence supports a role of herpesviruses (including HHV-6A and HHV-6B) as potential triggers. Interestingly, the infection by these viruses has been reported to impact the expression of microRNAs (miRNAs), short non-coding RNA sequences which have been suggested to be epigenetic factors modulating ME/CFS pathogenic mechanisms. Notably, the presence of circulating miRNAs in plasma has raised the possibility to use them as valuable biomarkers for distinguishing ME/CFS patients from healthy controls. Thus, this study aimed at determining the role of eight miRNAs, which were selected for their previous association with ME/CFS, as potential circulating biomarkers of the disease. Their presence was quantitatively evaluated in plasma from 40 ME/CFS patients and 20 healthy controls by specific Taqman assays, and the results showed that six out of the eight of the selected miRNAs were differently expressed in patients compared to controls; more specifically, five miRNAs were significantly upregulated (miR-127-3p, miR-142-5p, miR-143-3p, miR-150-5p, and miR-448), and one was downmodulated (miR-140-5p). MiRNA levels directly correlated with disease severity, whereas no significant correlations were observed with the plasma levels of seven pro-inflammatory cytokines or with the presence/load of HHV-6A/6B genome, as judged by specific PCR amplification. The results may open the way for further validation of miRNAs as new potential biomarkers in ME/CFS and increase the knowledge of the complex pathways involved in the ME/CFS development.

## 1. Introduction

Myalgic encephalomyelitis/chronic fatigue syndrome (ME/CFS) is a severe chronic disease that is characterized by unexplained debilitating fatigue, post-exertional malaise, localized or diffuse muscle pain, and sleep disturbances. The prevalence of ME/CFS in Europe ranges from 0.1% to 2.2% [1], although the estimate is affected by the poor knowledge of the disease, its challenging recognition, and the existence of different case definitions, based on different diagnostic criteria (among the commonly used ones, Centers for Disease Control & Prevention (CDC, 1994) [2], Canadian Consensus Criteria [3], London Criteria [4], International Consensus Criteria [5], or Institute of Medicine criteria) [6].

ME/CFS etiology is still unclarified, but it is recognized as a heterogeneous and multifactorial disease, and several factors have been hypothesized as triggers, including genetic predisposition, physical or emotional stress conditions, disruption of immunological processes, infection, and autoimmunity [7,8].

Several pieces of evidence support the role of human herpesviruses (HHVs), including HHV-6A, HHV-6B, HHV-7, and Epstein–Barr virus (EBV), as potential causative agents. To note, 49–93% of patients who developed ME/CFS disease reported an initial “flu-like” symptomatology, suggestive of undergoing viral infection or reactivation [9,10,11]. Consistently, HHV-6A/6B reactivation has been associated with the occurrence of ME/CFS clinical symptoms and higher levels of proinflammatory cytokines, including tumor necrosis factor (TNF)-α, interleukin (IL)-6, and IL-12 [12,13]. Saliva samples from ME/CFS patients were recently reported to harbor high loads of HHV-6 and HHV-7, which correlated with symptoms’ severity, thus supporting the hypothesis that HHV reactivation may have a role in ME/CFS pathogenesis and related immunological dysregulation [14]. Reactivation of HHV-6 was observed in the brain and neuronal tissues of ME/CFS patients, supporting the role of this virus in disease development [15].

However, despite increasing evidence implicating HHVs as potential etiological agents of ME/CFS, the underlying mechanisms are not clarified, and a few mechanistic hypotheses have been recently proposed [16,17]. HHV-6 reactivation was reported to induce mitochondrial fragmentation, decrease of ATP production, and an increase in reactive oxygen species, which are considered the key pathway in ME/CFS pathophysiology [16,18]. Other recent data evidenced a possible role of HHV-6A in altering germinal center activity and extrafollicular antibody responses by viral protein deoxyuridine triphosphate nucleotidohydrolase [17].

The lack of quantitative markers for ME/CFS diagnosis has stimulated several studies in the past 30 years, originally suggesting the role of immune response or dysfunction and trying to identify specific cytokines as biomarkers for the disease development [19,20]. Some data were, however, recognized as artifacts, such as those regarding the transforming growth factor β (TGFβ) marker, which, in fact, was related to the procedure of sample preparation rather than to genuine variation in serum concentration [20]. Other reports suggest that ME/CFS could be an autoimmune disease [8,21], and, indeed, increased plasmatic levels of antibodies against beta2-adrenergic receptors and muscarinic acetylcholine receptor 4 were found in ME/CFS patients compared to healthy controls [22].

More recently, growing interest has been given to microRNAs (miRNAs) as potential biomarkers in ME/CFS; however, so far, no miRNA has been validated for clinical diagnosis [23,24,25]. MiRNAs are short sequences (18–23 nucleotides) of non-coding RNA with essential roles in regulating gene expression at the post-transcriptional level, which is suggested to have a role in many pathological pathways. Specifically, circulating miR-124, miR-448, and miR-551b have been found to be differentially expressed in patients with rheumatoid arthritis (RA), systemic lupus erythematosus (SLE), Sjögren’s syndrome (SS), and ulcerative colitis, with respect to healthy controls, and thus have been suggested as biomarkers for autoimmune diseases [26]. High-throughput miRNome sequencing of plasma from ME/CFS subjects evidenced differential expression of miR-127-3p, miR-142-5p, and miR-143-3p compared to non-CFS controls [24]. In addition, the upregulation of miR-140-5p and miR-150-5p expression has been reported both in plasma and peripheral blood mononuclear cells (PBMCs) of ME/CFS subjects compared to healthy controls or associated with ME/CFS response to post-exertional malaise induction [23,25,27], and an influence of the nutritional status and gender of patients has been observed [27]. The main findings regarding the mentioned miRNAs and their association and role in ME/CFS disease are summarized in Table 1.

Interestingly, the infection by HHV-6 has been reported to affect the expression of miRNAs in different tissues and cellular types, particularly of those miRNAs also found to be deregulated in patients with autoimmune diseases [38,39,40].

Based on these observations, the aim of the present study was to determine the potential role of autoimmunity-associated miRNAs as biomarkers of ME/CFS. To this purpose, circulating miR-124-3p, miR-127-3p, miR-140-5p, miR-142-5p, miR-143-3p, miR-150-5p, miR-448, and miR-551b-3p were analyzed in plasma from 40 ME/CFS patients and 20 healthy controls (CTRs). Correlations were searched between miRNAs’ expression and disease severity, plasma pro-inflammatory cytokines, and HHV-6 infection/reactivation.

## 2. Results

### 2.1. Epidemiological and Clinical Features of ME/CFS and CTR Groups

A total of 60 subjects were recruited at the Rīga Stradiņš University outpatient clinic (Riga, Latvia), including 40 patients with clinical diagnosis of ME/CFS and 20 healthy subjects without medical history and symptoms of ME/CFS who were included as controls. Based on semi-structured interview questions created by Minnock et al. [41], ME/CFS patients were subdivided into three subgroups, according to the degree of disease severity (1, severe; 2, moderate; and 3, mild). Epidemiological characteristics (age and gender distribution) and ME/CFS severity of recruited patients were presented in Table 2. ME/CFS group included 9 men (30–69 years old) and 31 women (24–76 years old), with a mean age of 49.3 years. Overall, 5 patients out of 40 (12.5%) presented the most severe disease (grade 1), 22/40 (55%) were classified as severity grade 2 (moderate ME/CFS), and 13/40 (32.5%) presented the mildest degree of severity (grade 3). The control group included 4 men (19–38 years old) and 16 women (18–61 years old), with a mean age of 33.4 years. No statistically significant differences were observed in gender distribution between the ME/CFS and control groups (*p* = 1.00), whereas a statistically significant difference was observed for mean age (*p* < 0.0001), likely due to the low number or healthy subjects recruited.

### 2.2. Quantification of Pro-Inflammatory Cytokines in Plasma Samples

In order to correlate miRNA levels with the eventual inflammatory status of ME/CFS patients, the levels of seven of the most relevant cytokines involved in autoimmune diseases (IFN-γ, IL-17A, IL-2, IL-21, IL-23, IL-6, and TNF-α) were quantified in plasma samples from 39 patients and 20 controls, using the MILLIPLEX MAP Human High Sensitivity T Cell Panel—Immunology Multiplex Assay on Luminex 200 System. The results, which are summarized in Table 3, showed that five out of seven tested cytokines (IL-17A, IL-2, IL-21, IL-6, and TNF-α) were decreased in the ME/CFS group compared to the controls. By subdividing the ME/CFS group based on the severity of symptoms (1, severe; 2, moderate; and 3, mild), statistically significant differences were observed between the CTR group and ME/CFS-2 and ME/CFS-3 subgroups for IL-17A (*p* < 0.0001 and *p* = 0.0043, respectively), IL-2 (*p* < 0.0001 and *p* = 0.0004), IL-21 (*p* < 0.0001 and *p* = 0.0005), and IL-23 (*p* = 0.0013 and *p* = 0.0156). IL-6 levels were markedly reduced in ME/CFS-1 and -2 subgroups, compared to the controls (*p* = 0.0061 and *p* = 0.0308, respectively). Last, TNF-α resulted in being significantly decreased in ME/CFS grade 2 patients (*p* = 0.0038), compared to the CTR group. Interestingly, patients with the most severe disease (ME/CFS—grade 1) exhibited higher levels of all tested cytokines compared to subjects with moderate and mild disease, although the difference was not statistically significant. Although data regarding cytokine levels in ME/CFS disease are conflicting, a significant upward linear trend which correlated with ME/CFS severity has already been observed; however, in this case, cytokine levels in patients were found to be higher than in controls [19].

### 2.3. HHV-6A/B Presence and Load

In order to assess the presence and amount of HHV-6A/6B in the blood of ME/CFS subjects compared to controls, PBMCs were isolated from whole blood of patients and controls, and HHV-6A/B viral presence was analyzed by specific quantitative real-time PCR (RT-PCR), targeting the HHV-6 pol-gene. The results, reported in Figure 1, showed that a total of 9/40 patients (22.5%) resulted in being positive for the presence of HHV-6A/6B, with a mean viral load of 44,871.00 copies/10^6^ cells (range 5.71–403,454.00 copies/10^6^ cells). Among HHV-6-positive ME/CFS patients, two patients had severe symptoms (ME/CFS Subgroup 1; 2/5, 40%), six subjects showed moderate ME/CFS symptoms (Subgroup 2; 6/22, 27.3%), and one patient showed mild signs (Subgroup 3; 1/13, 33.3%), evidencing higher viral prevalence in the subgroup of patients with more severe symptoms; however, the differences among subgroups were not statistically significant (*p* = 0.22).

In the control group, only 2/20 subjects (10%) resulted in being positive for viral presence, with a mean viral load corresponding to 211.94 copies/10^6^ cells (range 174.80–249.08 copies/10^6^ cells). The differences detected between control and ME/CFS subjects, however, were not statistically significant, likely due to the low number of subjects included in the analysis and to the high variability of virus load in positive subjects.

### 2.4. miRNA Plasma Levels in ME/CFS Patients

The presence and amount of eight miRNAs, selected based on the literature data, were investigated in the plasma samples derived from ME/CFS patients and healthy controls by specific Taqman qPCR assays. The results showed that six miRNAs were differentially expressed in ME/CFS samples compared to the controls (Figure 2a). Among them, in particular, miR-142, miR-150, and miR-448 were increased in ME/CFS plasma specimens compared to the controls (*p* = 0.02, *p* = 0.03, and *p* < 0.0001 respectively), while miR-140 resulted in being significantly downmodulated in the ME/CFS group, as compared to the controls (*p* = 0.007).

By stratifying ME/CFS patients according to symptoms severity (1, severe; 2, moderate; and 3, mild), very different levels of circulating miRNAs were detected within the three subgroups (Figure 2b). Specifically, miR-124 and miR-142 appeared to be increased in the subgroups with a higher level of disease severity, although differences between the patient groups and control group were not statistically significant. Similarly, the downmodulation of miR-140 correlated with the severity of symptoms, although the decrease resulted in being significant only for the ME/CFS Subgroup 2 compared to the controls (*p* = 0.006). Overall, increased levels of plasmatic miRNAs in the ME/CFS group were more evident by stratifying patients according to severity, showing a positive correlation between abundance of miRNAs and more severe symptoms. This was evident for miR-448, which was overexpressed in all symptoms’ subgroups, as compared to healthy controls, with increasing values correlating with ME/CFS severity, (*p* < 0.0001, *p* = 0.001, and *p* = 0.0015, for Subgroups 1, 2, and 3, respectively). Statistically significant increases were also detected for miR-127 (*p* = 0.0019), miR-143 (*p* = 0.5), and mir-150 (*p* = 0.0007) in ME/CFS Subgroup 1 when compared to the controls.

### 2.5. Gene Pathways Analysis

To investigate the possible pathways affected by the miRNAs resulting from dysregulation in the ME/CFS patients, gene pathways and network analyses were performed by using the MIENTURNET web tool [42]. Potential genes regulated by altered miRNAs were computationally predicted based on the miRTarBase reference database. The network of experimentally validated miRNA–target interactions identified by the enrichment analysis was built while considering both strong and weak experimental methods (Figure 3a).

Most predicted interactions involved genes encoding the Zinc Finger Protein 426 (ZNF426), Matrix Metallopeptidase 13 and 14 (MMP13 and MMP14), Signal Transducer and Activator of Transcription 1 (STAT1), Gene-Spi-C Transcription Factor (SPIC), Single-Strand-Selective Monofunctional Uracil-DNA Glycosylase 1 (SMUG1), SMAD Family Member 3 (SMAD3), PR/SET Domain 1 (PRDM1), and Peptidylprolyl Isomerase E (PPIE).

Functional enrichment analysis, performed by considering WikiPathways database to evidence eventual specific ME/CFS-associated genes (Figure 3c), showed that pathways regulated by altered miRNAs were involved in extracellular matrix remodeling (matrix metalloproteinases and Focal Adhesion), cytokines-mediated signaling pathway (IL-6, Oncostatin M, IL-7, and TGF-β), apoptosis, cell-cycle regulation via the P13K-m-TOR signaling pathway, immune response to microbial infection (NOD pathway), and senescence and autophagy in pathologic conditions.

No direct positive correlation was detected by the Spearman analysis between any of the analyzed miRNAs, inflammatory cytokines, and patients’ HHV-6A/B positivity, while some inverse correlations were observed between miR-448 and all assayed cytokines, except for IL-17A and TNFα (r range = −0.259/−0.438; *p* < 0.05) (Appendix A).

## 3. Discussion

Currently, the only available diagnostic methods for ME/CFS are clinical, based on symptom-related criteria, which also leads to undiagnosed or misclassified cases due to symptom heterogeneity. Thus, specific biomarkers to be used for ME/CFS diagnosis are urgently needed.

Recently, different expressions of circulating miRNAs have been reported in ME/CFS patients compared to healthy subjects, suggesting their use as a signature to discriminate ME/CFS disease. In parallel, HHV-6A/B infection was suggested to be a trigger of ME/CFS onset and/or progression, and in vitro infection by such viruses was reportedly shown to induce alterations of miRNA expression in infected cells, possibly correlated with inflammation, cell apoptosis, and fibrosis [12,13,14,15,16,17,38,39,40,43].

Thus, this study aimed to assess, for the first time in the Latvian population, the diagnostic value of miRNA signatures in distinguishing patients with ME/CFS from healthy controls. Meanwhile, the status of patients with regard to HHV-6A/B positivity and concentration of plasma proinflammatory cytokines was analyzed in order to evidence any eventual correlation with miRNA deregulation.

The expression levels of eight miRNAs, selected based on the literature data [24,25,26], were quantified by specific RT-qPCR in plasma samples derived from 40 subjects with ME/CFS diagnosis and 20 healthy controls. The results evidenced significant upregulation of miR-142-5p, miR-150-5p, and miR-448 in ME/CFS group, compared to the controls, confirming previously reported data [24,25,26]. Moreover, the increases were more evident and statistically significant in the subgroups of ME/CFS patients showing more severe symptoms (Subgroups 1 and 2), evidencing a direct correlation between miRNA amount and symptoms severity. In addition, two more miRNAs were significantly upregulated in severe ME/CFS patients as compared to the controls: miR-127-3p and miR-143-3p, suggesting that those miRNAs may be used as markers for severe disease. In our study, miR-140-5p was the only miRNA significantly downmodulated in ME/CFS subjects compared to healthy individuals (*p* = 0.07).

Our results are in line with those reported in other ME/CFS patients’ cohorts. Nineteen miRNAs were reported to be differentially expressed at the plasma level in ME/CFS patients compared to non-fatigued controls, and significant upregulations of miR-127-3p, miR-142-5p, and miR-143-3p were detected [24]. More recently, the plasma levels of miR-127-3p, miR-140-5p, and miR-150-5p were found to be increased in ME/CFS patients compared to the controls after post-exertional stress challenge, suggesting miR-127 and miR-140 as biomarkers to discriminate between patients suffering from ME/CFS and fibromyalgia [25,29]. In addition, PBMCs of ME/CFS patients were shown to harbor increased levels of miR-140-5p [23] and miR-150-5p in response to exercise [27].

From a mechanistical point of view, miR-127-3p expression has been shown to inhibit the expression of IL-10 via regulation of the B-Cell Lymphoma 6 Protein (*BCL6*) gene [30]. IL-10, an important anti-inflammatory cytokine that suppresses Th_1_-related responses, has been consistently observed to be significantly reduced in the cerebrospinal fluid of ME/CFS patients in comparison to the controls [44]. In addition, miR-127 upregulation has been reported to inhibit cell proliferation and induce apoptosis [31,45].

Among the miRNAs found to be upregulated in ME/CFS patients, miR-142-5p has been reported as being overexpressed in most diseases linked to immunological disorders [46], and miR-143-3p has been identified as a neutrophil-specific miRNA [47] that is upregulated during increased erythropoiesis in polycythemia [48]. Of note, both miR-142-5p and miR-143-3p have been proved to interact with TGF-β1 through various mechanisms and modulate fibrotic processes [33,34,49].

The upregulation observed in our cohort of ME/CFS patients is in line with what was previously reported, showing that the overexpression of miR-150-5p in ME/CFS correlates with higher post-exertional malaise scores and symptom severity of patients [25,29]. This lymphopoietic-specific miRNA regulates many genes involved in the differentiation and proliferation of immune cells [35,50], and its pivotal role in autoimmune disease such as autoimmune pancreatitis, systemic SLE, primary Sjögren’s syndrome, and multiple sclerosis has been pointed out [51,52,53].

Similarly, plasmatic miR-448 has been already suggested as a valuable biomarker for distinguishing patients affected by autoimmune diseases, including RA, SLE, SS, ulcerative colitis, and MS, from healthy controls [26,54]. Based on our knowledge, our results evidence, for the first time, the potential diagnostic value of miR-448 also in ME/CFS.

Mixed results were instead obtained regarding miR-140-5p, whose expression was found to be increased in some ME/CFS cohorts [25,29] and decreased in our group of patients. This miRNA regulates different pathways, including cell proliferation, apoptosis, and inflammatory cascades [32]; it is implicated in immune-related disorders, through TLR4/NF-κB signaling modulation, and has been found to be downregulated in several neoplastic diseases [32].

To have a comprehensive view of the pathways regulated by the miRNAs that were found to be altered in our ME/CFS cohort, we performed gene pathways and functional enriched analyses. One of the principal target genes for number of interactions was *ZNF426*, encoding a zinc finger transcriptional repressor that modulates the reactivation of Kaposi’s sarcoma-associated herpesvirus (HHV-8) [55], suggesting a potential role of this target also in other human herpesviruses. Other identified target genes included *MMP13* and *MMP14* of the metalloproteinase family, involved in tissue remodeling and cartilage degradation, which are activated in non-pathological post-exercise conditions but can be associated with pathologic processes, including tumor invasion and arthritis [56,57,58,59]. Further predicted genes included *STAT1*, a gene coding cytokine-induced factors, including interferons (IFNs), EGF, PDGF, and IL-6 [60,61]. STAT1 protein has a crucial role in regulating immune responses to viral, fungal, and mycobacterial pathogens, and its mutation has been associated with pathological immunodeficiency [62]. In addition, STAT1 has been recognized to be involved in chronic fatigue and immune deficiency syndrome (CFIDS) and can mediate mitochondrial dysfunction by ROS and disruption of ATP production [63]. Another recognized target gene was *SPIC*, which controls the development of red pulp macrophages, which are essential for iron homeostasis and the recycling of red blood cells (RBCs) [64]. The alteration of this pathway could have a role in the phenotypic alteration and decrease of deformability affecting RBCs of ME/CFS patients, as compared to healthy individuals [65]. Last, other genes potentially affected by investigated miRNAs are involved in the regulation of innate and adaptive immune tissue-resident lymphocyte T cells via the β-IFN pathway (*PRDM1* gene), modulation of the TGF-β signaling pathway (*SMAD3* gene), DNA repair mechanisms (*SMUG1* gene), and protein folding (*PPIE* gene) [66,67,68,69].

By contrast, the analysis of the inflammatory status of our ME/CFS cohort, by quantifying the plasmatic levels of seven cytokines, did not reveal specific signatures associated with ME/CFS or correlated with disease severity. Rather, we found out that IL-2, IL-21, IL-6, IL-17A, and TNF-α were significantly decreased in patients compared to controls. Although these cytokines have a significant proinflammatory role, contradictory data have been published regarding their role in ME/CFS. In fact, by analyzing whether a signature of 51 serum cytokines could be associated with ME/CFS and correlated with disease severity, only TGF-β and resistin appeared to be significantly altered in patients compared to controls [19]. Notably, despite the fact that resistin is known to have a significant proinflammatory role [70], it was decreased in ME/CFS subjects. With regard to IL-2, some studies reported increased levels in CFS patients compared to controls, whereas, in others, decreased IL-2 levels or no difference was reported between patients and control groups [71]. In a similar way, decreased IL-6 levels were reported in mild/moderate ME/CFS patients compared with both healthy controls and severe ME/CFS patients [72].

The observed results may be due to the use of nonsteroidal anti-inflammatory drugs and benzodiazepines in ME/CFS patients that could lower the concentration of inflammatory cytokines in comparison to the healthy controls, as also reported in the scientific literature regarding these medications [73]. Thus, taking into account that, in our study, cytokine levels were measured only in 40 patients, it is not possible to draw general conclusions about cytokine signatures related to disease severity in ME/CFS. Similarly, no significant direct correlation was observed between altered miRNAs and inflammatory cytokines, while a weak inverse correlation was found between miR-448 and most of the tested cytokines (IFN-γ, IL-2, IL-21, IL-23, and IL-6). However, this aspect may deserve further investigation and validation in a larger cohort of patients.

The literature data support the role of HHV-6A/B as a potential trigger of ME/CFS, highlighting the association between HHV-6A/B infection and ME/CFS development [9,22]. In our study cohort, the percentage of HHV-6A/B positive subjects was doubled in the ME/CFS group compared to the controls (22.5% vs. 10%), and the viral load in the PBMCs of ME/CFS patients was remarkably higher than in the controls (mean of 44,871.00 copies/10^6^ cells vs. 211.9 copies/10^6^ cells); however, the differences were not statistically significant, likely due to the low number of subjects included. In this regard, given that these results may be clues of HHV-6A/B’s involvement in the disease, enlarging the cohort of patients and controls will be important to confirm, in a statistically significant way, HHV-6A/B as one of the potential biomarkers for ME/CFS diagnosis.

Limitations of this study included the low number of enrolled subjects and the differences in mean age between patients and the control group, which should be acknowledged and taken into account. Future studies should thus expand the number of individuals and find adequate control cohorts (with matched age and gender) to confirm the role of ME/CFS-associated miRNAs as potential diagnostic biomarkers in order to further investigate their possible correlation with inflammatory status and viral infection in ME/CFS patients and deepen our understanding of the mechanisms by which they may induce viral and host gene regulation during the disease onset and progression.

## 4. Materials and Methods

### 4.1. Study Population

A total of 60 subjects were recruited at the Rīga Stradiņš University outpatient clinic. The ME/CFS group included 40 patients with ME/CFS clinical diagnosis based on the Fukuda criteria [2], and based on adapted semi-structured interview questions created by Minnock et al. [41], patients were stratified into three subgroups according to disease severity: 1, severe; 2, moderate; 3, mild. The control group included 20 subjects with no medical history or symptoms of chronic fatigue syndrome. ME/CFS and control groups were matched in terms of sex distribution; however, due to a lack of healthy individuals involved in this study, age equality was not achieved. The study was conducted in accordance with the Declaration of Helsinki and obtained approval by the Ethical Committee of Rīga Stradiņš University (Ethical code Nr.6-1/05/33 and date of approval 30 April 2020). Prior to recruitment, informed consent was obtained from all subjects involved in the study.

### 4.2. Samples Collection

Ten milliliters of peripheral blood was collected from each participant in EDTA-treated tubes and immediately transported to the laboratory for processing. PBMCs were isolated by Ficoll-Hypaque gradient, as previously reported [74]. PBMCs (5 × 10^5^ aliquots) in Trizol and plasma fractions were frozen and stored at −80 °C until the analysis.

### 4.3. Cytokines Evaluation

Detection of IL-2, IL-17, IL-6, IL-21, IL-23, TNF-α, and IFN-γ levels in plasma samples of 39 ME/CFS patients and 20 healthy controls was carried out with Luminex 200 Instrument System, using a commercially available kit (MILLIPLEX MAP Human High Sensitivity T Cell Panel—Immunology Multiplex Assay), according to the manufacturer’s protocol.

### 4.4. DNA Extraction and Analyses of HHV-6A/B Presence

DNA was isolated from PBMC samples via the phenol–chloroform extraction method; DNA quality and concentration were evaluated by spectrophotometric reading, using Nanodrop instrument (ThermoFisher Scientific, Waltham, MA, USA, NanoDrop 1000); and β-globin PCR was also performed to determine DNA quality, as previously described [75]. HHV-6 Real-TM Quant amplification test (Sacace Biotechnologies; Como, Italy) was used for quantitative detection of HHV-6A/B in DNA of PBMCs. DNA was amplified using real-time amplification with fluorescent reporter dye probes specific for pol-gene of HHV-6A/B and internal control.

### 4.5. RNA Extraction and miRNA Analysis

Plasma samples were thawed on ice and additionally centrifuged for 10 min at 15,000× *g* at 4 °C to remove residual cellular debris. Total RNA, including miRNA fraction, was extracted from plasma samples, using the MagMax mirVana Total RNA isolation kit (ThermoFisher Scientific, Waltham, MA, USA), based on magnetic-bead technology, following the manufacturer’s instructions. Synthetic ath-miR-159a (ThermoFisher Scientific, Waltham, MA, USA) was combined with plasma samples during the lysis step as spike-in control to monitor the extraction efficiency. After reverse transcription by the TaqMan advanced miRNA cDNA synthesis kit (ThermoFisher Scientific, Waltham, MA, USA), cDNA templates were analyzed by the TaqMan Advanced miRNA assays (ThermoFisher Scientific, Waltham, MA, USA). Individual target miRNAs included hsa-miR-448, hsa-miR-124-3p, hsa-miR-551b-3p, hsa-miR-127-3p, hsa-miR-142-5p, hsa-miR-143-3p, hsa-miR-140-5p, and hsa-miR-150-5p. In addition, two endogenous controls (hsa-miR-361-5p and hsa-miR-186-5p) and an exogenous control (spike-in ath-miR-159a) were tested. All real-time qPCR reactions were performed using the QuantStudio5 instrument (Applied Biosystem, Waltham, MA, USA).

The expression levels of selected miRNAs were quantified by using the ∆∆Ct method. Briefly, ∆Ct values were obtained for each sample, subtracting the Ct value of each miRNA from that of the exogenous control, ath-miR-159a. Secondly, ∆∆Ct values were calculated for every sample, as the difference between the normalized ∆Ct value and the average ∆Ct values of controls. Lastly, relative miRNA expression values were calculated and expressed as 2^−∆∆Ct^. To avoid heavily skewed data, Log-transformed 2^−∆∆Ct^ values were used for statistical analysis.

### 4.6. Gene Pathways and Functional Enriched Analysis

The potential target genes of altered miRNAs were identified by using the MIENTURNET web tool, based on experimentally validated miRNA-target interactions collected in the miRTarBase reference database [42]. For each gene in the chosen database (TargetScan or miRTarBase), the hypergeometric test was used to calculate the significance (*p*-value < 0.05). The network of miRNA–target interactions identified by the enrichment analysis was built considering both strong and weak experimental methods. The thresholds for the minimum number of miRNA–target interactions and for the adjusted *p*-values, obtained by using the Benjamini-Hochberg (False Discovery Rate, FDR) procedure for multiple testing, were set as 2 and 1, respectively (default settings). A functional enrichment analysis of target genes of selected miRNAs was performed while considering the following annotation databases: KEGG, REACTOME, WikiPathways, Disease Ontology. The results obtained from WikiPathways are shown.

### 4.7. Statistical Analysis

Statistical analysis and graphical representations were performed using GraphPad Prism 5.03 software (GraphPad Software, San Diego, CA, USA). Unpaired Student’s *t*-test was used to compare miRNAs’ relative expression values between the control group and ME/CFS patients’ group. One-way ANOVA, followed by Bonferroni’s multiple comparisons test and Spearman Correlation analysis, was applied to investigate the correlation between presence/number of miRNAs and patients’ variables (disease severity and levels of inflammatory cytokines). Values of *p* ≤ 0.05 were considered statistically significant. The statistical power of the study was evaluated during the study design by using G*Power 3.1 free software (Heinrich Heine University Düsseldorf, Düsseldorf, Germany, HHU), error probability α =0.05, and power level pβ = 0.8.

## 5. Conclusions

ME/CFS disease has a yet unclarified etiology and suffers from the lack of distinctive diagnostic biomarkers. Our findings show that specific circulating miRNAs (miR-127-3p, miR-140-5p, miR-142-5p, miR-143-3p, miR-150-5p, and miR-448), are differentially expressed in the plasma of ME/CFS patients compared to healthy controls, correlating with disease severity. Of note, these miRNAs are involved in immune and inflammatory response pathways, suggesting their role in ME/CFS pathogenesis and their possible use as useful blood markers to help identify ME/CFS patients. In addition, the collected data suggest a possible involvement of HHV-6A/B infection as a potential environmental ME/CFS trigger, though the differences between ME/CFS patients and controls did not result in being statistically significant.

## Figures and Tables

**Figure 1 ijms-24-10582-f001:**
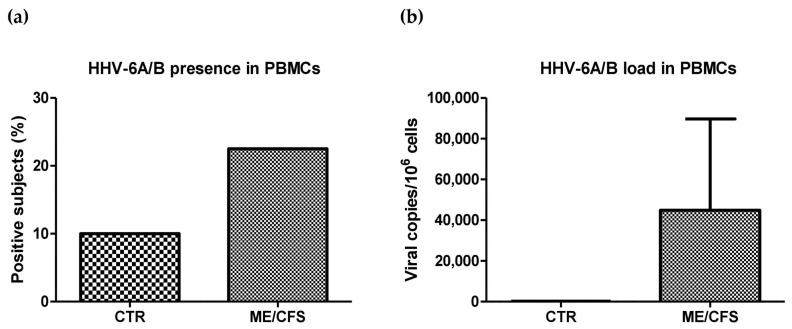
HHV-6A/B presence and load in enrolled subjects. (**a**) HHV-6A/B-positive subjects in control (CTR) and ME/CFS groups; results are expressed as percentage of positive individuals on the total enrolled subjects. (**b**) HHV-6A/B load in CTR and ME/CFS groups; results are expressed as mean viral genome copy number per 10^6^ PBMCs ± S.E. (standard error).

**Figure 2 ijms-24-10582-f002:**
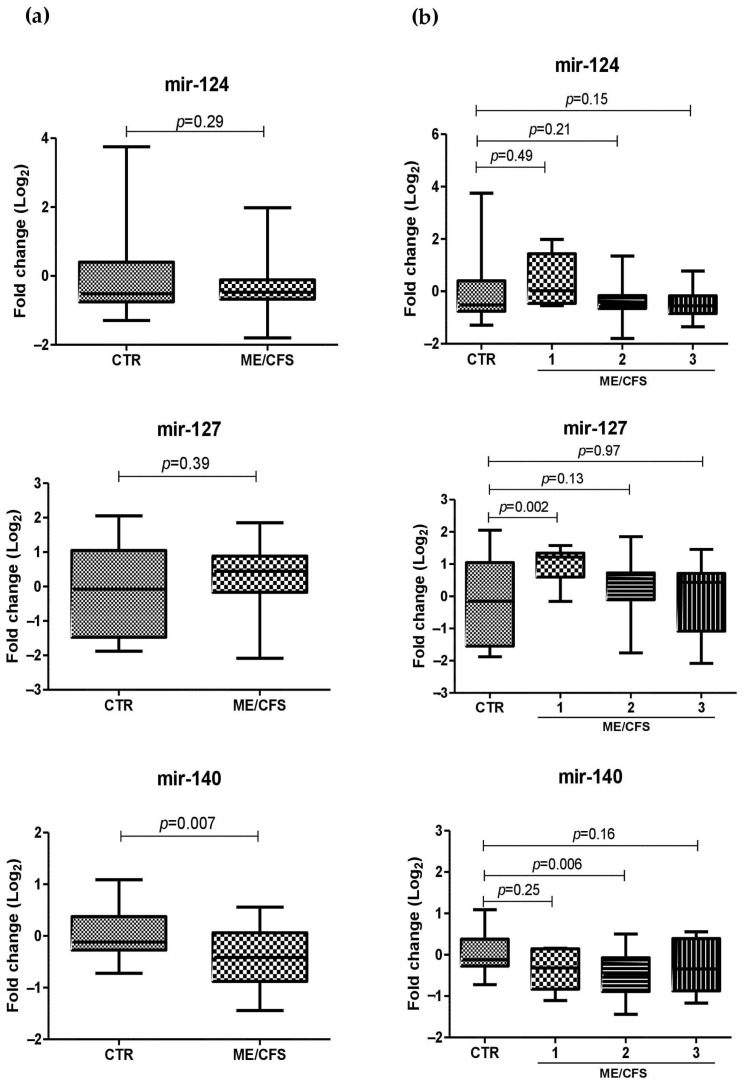
Plasma miRNA levels in control (CTR) and ME/CFS subjects. (**a**) Comparison between whole CTR and ME/CFS groups. (**b**) Comparison between CTR group and ME/CFS subgroups, subdivided for symptoms severity (1, severe; 2, moderate; and 3, mild). All results are expressed as fold change (log_10_ values) detected in ME/CFS subjects compared to the controls. Results are depicted as boxplots with Whiskers; the median line, interquartile range, and min-max values for each group are shown. Statistical significance is also shown, as obtained by the unpaired *t*-test and ANOVA test for multiple comparisons.

**Figure 3 ijms-24-10582-f003:**
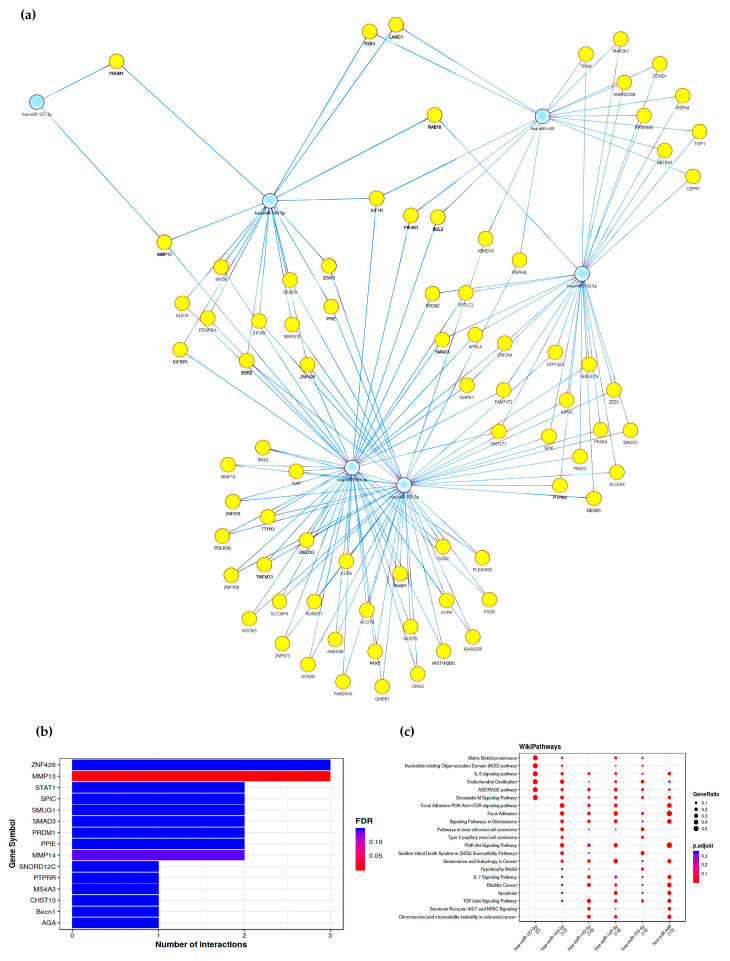
Predicted gene pathways of altered miRNAs in ME/CFS plasma. (**a**) Potential pathways affected by miR-127, miR-140, miR-142, miR-143, miR-150, and miR-448; miRNAs are represented in blue circles, target genes are represented in yellow circles, each blue line represents a miRNA-gene interaction. (**b**) Main target genes involved in detected pathways; the top ten target genes resulted from the analysis and the number of miRNAs targeting them are presented. The color code reflects the adjusted p-values for multiple testing (False Discovery Rate, FDR), increasing from red to blue. (**c**) Functional enrichment analysis based on WikiPathways; colors from red to blue of the dots represent the adjusted p-values (FDR), whereas the size of the dots represents the gene ratio (n° of miRNA targets found in each category/n° of total miRNA targets). Analyses and graphical representations were obtained using the MIENTURNET tool [42].

**Table 1 ijms-24-10582-t001:** Association of selected miRNAs with ME/CFS and potential role in the disease.

Target miRNAs	Association and Potential Role in ME/CFS	References
miR-124-3p	Significantly decreased in RA, SLE, SS, and UC subjects, compared to healthy controls; suggested as biomarker for systemic autoimmune diseases: AUC 0.9 (95% CI 0.833–0.967), 76.5% specificity and 91.3% sensitivity. It regulates autoimmune inflammation.	Jin F. et al., 2018 [26] Ponomarev E.D. et al., 2011 [28]
miR-127-3p	Upregulated in plasma of ME/CFS subjects compared to non-fatigued controls. Increased in PEM. Identified as a potential biomarker to distinguish ME/CFS disease from fibromyalgia. It suppresses IL-10 response, inhibits cell proliferation, and induces cell apoptosis.	Brenu E.W et al., 2014 [24] Nepotchatykh, E. et al., 2020 [25] Nepotchatykh, E. et al., 2023 [29] Saito Y. et al., 2006 [30] Wei G. et al., 2023 [31]
miR-140-5p	Increased in plasma of ME/CFS subjects. Identified as a potential biomarker to distinguish ME/CFS from fibromyalgia. Upregulated in PBMCs of ME/CFS patients. It suppresses IL-10 response and modulates T-cell differentiation and proliferation of immune cells.	Nepotchatykh, E. et al., 2020 [25] Nepotchatykh, E. et al., 2023 [29] Almenar-Pérez E. et al., 2020 [23] Ghafouri-Fard S. et al., 2021 [32]
miR-142-5p	Upregulated in plasma of ME/CFS subjects compared to non-fatigued controls. It modulates differentiation/proliferation of immune cells and interaction with TGF-β1 pathway.	Brenu E.W. et al., 2014 [24] Wang Z. et al., 2020 [33]
miR-143-3p	Upregulated in plasma of ME/CFS subjects compared to non-fatigued controls. It modulates differentiation/proliferation of immune cells and interaction with TGF-β1 pathway.	Brenu E.W. et al., 2014 [24] Cheng W. et al., 2016 [34]
miR-150-5p	Higher PEM scores and increased symptom severity of ME/CFS patients. Upregulated in PBMCs of ME/CFS patients in response to exercise. It modulates differentiation/proliferation of immune cells.	Nepotchatykh, E. et al., 2020 [25] Nepotchatykh, E. et al., 2023 [29] Cheema A.K. et al., 2020 [27] Ménoret A. et al., 2023 [35]
miR-448	Significantly increased in RA, SLE, SS, UC, compared to healthy controls. Suggested as biomarkers for systemic autoimmune diseases: AUC 0.91 (95% CI 0.85–0.97), 82.4% specificity and 91.3% sensitivity. Mainly studied in cancer-related pathways.	Jin F. et al., 2018 [26] Liao Z.B. et al., 2019 [36]
miR-551b-3p	Significantly increased in RA, SLE, SS, and UC subjects compared to healthy controls; suggested as biomarkers for systemic autoimmune diseases: AUC 0.850 (95% CI 0.769–0.932), 73.5% specificity, and 88.4% sensitivity. It regulates inflammatory response.	Jin F. et al., 2018 [26] Zhang Y. et al., 2018 [37]

RA, rheumatoid arthritis; SLE, systemic lupus erythematosus; SS, Sjögren’s syndrome; UC, ulcerative colitis; AUC, area under the ROC curve; ME/CFS, myalgic encephalomyelitis/chronic fatigue syndrome; PEM, post-exertional malaise; PBMCs, peripheral blood mononuclear cells.

**Table 2 ijms-24-10582-t002:** Epidemiological features of ME/CFS patients and healthy controls.

Group	N	Age Mean ± SE ^1^	Gender	ME/CFS Severity ^2^		
				1	2	3
ME/CFS	40	49.30 ± 2.234	F: 31 (77.5%) M: 9 (22.5%)	5 (12.5%)	22 (55%)	13 (32.5%)
Controls	20	33.40 ± 2.634	F: 16 (80%) M: 4 (40%)	-	-	-
*p* value		0.0001	1.00 (n.s)			

^1^ Standard error. ^2^ ME/CFS symptoms severity based on the semi-structured interview questions created by Minnock et al. (1, severe; 2, moderate; and 3, mild). n.s., not significant.

**Table 3 ijms-24-10582-t003:** Plasma cytokine quantification ^1^.

Cytokine	Group (Subjects n°)								
	Controls (20)	ME/CFS (39)	*p*-Value ^2^	ME/CFS Grade 1 (5)	*p*-Value ^2^	ME/CFS Grade 2 (22)	*p*-Value ^2^	ME/CFS Grade 3 (13)	*p*-Value ^2^
**IFN-γ**	649.90 ± 61.33	507.7 ± 49.98	0.1560	675.8 ± 92.42	0.2890	429.2 ± 77.06	0.1634	572.3 ± 79.80	0.5089
**IL-17A**	103.80 ± 12.40	18.80 ± 5.54	0.0003	20.47 ± 20.43	0.1568	16.66 ± 5.55	<0.0001	19.48 ± 11.76	0.0043
**IL-2**	30.40 ± 3.99	4.52 ± 1.42	0.0001	7.26 ± 7.17	0.1989	4.28 ± 1.10	<0.0001	4.40 ± 2.63	0.0004
**IL-21**	43.97 ± 5.94	9.07 ± 2.13	0.0001	10.46 ± 9.20	0.1065	8.68 ± 2.18	<0.0001	9.64 ± 3.99	0.0005
**IL-23**	1.602 ± 235.00	407.5 ± 114.5	0.1065	527.5 ± 620.3	0.6728	403.2 ± 100.3	0.0013	377.4 ± 194.3	0.0156
**IL-6**	16.34 ± 8.14	0.90 ± 2.13	0.0270	0.95 ± 0.23	0.0061	0.90 ± 2.96	0.0308	0.74 ± 4.31	0.0588
**TNF-α**	45.16 ± 4.06	12.35 ± 3.42	0.0052	13.44 ± 11.27	0.6397	12.35 ±4.04	0.0038	11.61 ± 6.89	0.0587

^1^ Results are expressed as median cytokine concentration (pg/mL) ± standard error. ^2^ *p*-value obtained by comparing all ME/CFS patients or ME/CFS subgroups (1, severe; 2, moderate; and 3, mild) with the group of controls.

## Data Availability

All data generated by the study are included in the manuscript or in its Appendix A.

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
