# Peer review of "Circulating miRNAs Expression in Myalgic Encephalomyelitis/Chronic Fatigue Syndrome"

_ijms, 2023, doi:10.3390/ijms241310582_

Round 1
Reviewer 1 Report
Reviewer Comments
In the present research article on “Circulating miRNAs expression in myalgic encephalomyelitis/chronic fatigue syndrome”, the authors have conducted experimental work on determining the role of eight miRNAs, selected for their previous association with ME/CFS, as potential circulating biomarkers of the disease. In plasma from 40 ME/CFS patients and 20 healthy controls, their presence was quantitatively assessed using specific Taqman assays. The results revealed that six of the eight selected miRNAs were differentially expressed in patients compared to controls; more specifically, five miRNAs were significantly up-regulated. The finding of this research work implied that these miRNAs have the potential as a biomarker in the diagnosis of ME/CFS. Overall, the research work is interesting however due to the small sample size the results are not showing any significant correlation between the level of plasma cytokines and miRNA.
The paper will be accepted only after the incorporation of the response to suggested comments.
Scientific comments
1. Did the authors calculate the power of the study? If yes, please mention it in the revised MS.
2. In the introduction part or in the discussion part add a table showing the role of these eight miRNAs in Myalgic encephalomyelitis/chronic fatigue syndrome so it is easy to understand the importance of these miRNAs as a biomarker.
3. The authors have quantified the level of the most relevant cytokines involved in autoimmune diseases to understand the inflammatory status of ME/CFS patients in order to correlate miRNA levels. However, in the discussion section, you have not given a correlation of all selected cytokines with selected miRNAs.
4. In the result section you have mentioned (lines 140-143), “Interestingly, patients with the most severe disease …. statistically significant”, Discuss this result outcome in detail. It is not understandable. Is this result supported by other previous studies? Discuss with specific reference.
5. In the discussion section, Line 320-321, “Rather, we found out that IL-2, ……. patients compared to controls”, discuss this in detail with some references.
6. Either add a table or figure showing a correlation between cytokines and miRNA with respect to Myalgic encephalomyelitis/chronic fatigue syndrome.
7. In the abstract and introduction, the authors have mentioned the “role of herpesviruses (including HHV-6A and HHV-6B) as potential triggers of ME/CFS however the result is not showing any significant correlation. And you have discussed it might be due to the small size of the sample. However, in the discussion section in lines 339-341, you mentioned “Instead, our results support the role of HHV-6A/B as a potential trigger of ME/CFS, confirming the association between HHV-6A/B infection and ME/CFS development previously reported”. This line looks controversial with your result. Please check and discuss it more clearly.
8. These eight miRNAs are already discussed in various research papers. How are your research outcomes increase the knowledge in the same?
Reviewer 2 Report
Excellent study that will advance patient care and understanding of these conditions. Highly recommend publication. Ok as is
Round 2
Reviewer 1 Report
Reviewer Comments
All the scientific comments given in the previous report are very well responded to by the authors. The result and conclusion are updated accordingly.
The paper can be accepted in its present form.